# Modulation of Trans-Synaptic Neurexin–Neuroligin Interaction in Pathological Pain

**DOI:** 10.3390/cells11121940

**Published:** 2022-06-16

**Authors:** Huili Li, Ruijuan Guo, Yun Guan, Junfa Li, Yun Wang

**Affiliations:** 1Department of Anesthesiology, Beijing Chaoyang Hospital, Capital Medical University, Beijing 100020, China; lily@mail.ccmu.edu.cn; 2Department of Anesthesiology, Beijing Friendship Hospital, Capital Medical University, Beijing 100030, China; guoguo12015@163.com; 3Department of Anesthesiology and Critical Care Medicine, Johns Hopkins University School of Medicine, Baltimore, MD 21205, USA; yguan1@jhmi.edu; 4Department of Neurobiology, Capital Medical University, Beijing 100069, China; junfali@ccmu.edu.cn

**Keywords:** neurexin, neuroligin, pain, synapse, plasticity

## Abstract

Synapses serve as the interface for the transmission of information between neurons in the central nervous system. The structural and functional characteristics of synapses are highly dynamic, exhibiting extensive plasticity that is shaped by neural activity and regulated primarily by trans-synaptic cell-adhesion molecules (CAMs). Prototypical trans-synaptic CAMs, such as neurexins (Nrxs) and neuroligins (Nlgs), directly regulate the assembly of presynaptic and postsynaptic molecules, including synaptic vesicles, active zone proteins, and receptors. Therefore, the trans-synaptic adhesion mechanisms mediated by Nrx–Nlg interaction can contribute to a range of synaptopathies in the context of pathological pain and other neurological disorders. The present review provides an overview of the current understanding of the roles of Nrx–Nlg interaction in the regulation of trans-synaptic connections, with a specific focus on Nrx and Nlg structures, the dynamic shaping of synaptic function, and the dysregulation of Nrx–Nlg in pathological pain. Additionally, we discuss a range of proteins capable of modulating Nrx–Nlg interactions at the synaptic cleft, with the objective of providing a foundation to guide the future development of novel therapeutic agents for managing pathological pain.

## 1. Introduction

Synapses serve as the interface that mediates inter-neuronal communication within the nervous system [1]. To maintain the high degree of plasticity necessary to respond to a variety of stimuli, synapses must be capable of dynamically altering the expression of specific postsynaptic receptors and channels and controlling the balance between the production and release of neurotransmitter-containing presynaptic vesicles [2,3]. Through the trans-synaptic regulation of these processes and associated synaptic architecture, synapses can fine-tune the transmission of neuronal signals.

Several candidate regulators of synaptosome architecture have been identified, with prototypical trans-synaptic cell-adhesion molecules (CAMs), such as neurexins (Nrxs) and neuroligins (Nlgs), playing a central role in this context [4]. Thus, regulating the expression and function of Nrxs and Nlgs may profoundly influence the processing of trans-synaptic signals, and subsequent dysfunctions in trans-synaptic regulatory activity may lead to synaptopathies in a variety of neurological disorders, including pathological pain [5,6].

Herein, we provide a detailed review of preclinical evidence regarding the importance of Nrx–Nlg in the context of the trans-synaptic dynamic modulation of the structural and functional characteristics of neuronal synapses. In particular, we elucidate the mechanisms by which they contribute to pathological pain conditions. Moreover, we summarize a range of proteins that can modulate Nrx–Nlg trans-synaptic interactions, underscoring a need to explore the mechanisms controlling the Nrx–Nlg interaction and their roles in the trans-synaptic regulatory network to clarify whether these regulators may be potential new targets for pain treatment.

## 2. The Structural Characteristics of Neurexins and Neuroligins

### 2.1. The Structural Diversity of Neurexins

Nrxs are evolutionarily conserved presynaptic CAMs encoded by three mammalian genes (*Nrx-1*, *Nrx-2*, *Nrx-3*), each of which contains two promoters, leading to the generation of longer and shorter α-Nrx and β-Nrx pre-mRNAs, respectively. In mice, a third very short γ-isoform of Nrx-1 is produced via an additional internal promoter [7]. Structurally, α-Nrxs consist of six laminin/neurexin/sex-hormone (LNS) globular domains and three interspersed extracellular epidermal growth factor (EGF)-like repeats, with these alternating structures serving to tether proteins to the cell surface through a rigid and highly O-linked glycosylated stalk and a transmembrane domain. These α-Nrxs also exhibit a short cytoplasmic tail region consisting of cytoskeletal adapter protein interaction sites and a C-terminal PSD-95, DLG1, ZO-1 (PDZ) binding motif. The shorter β-Nrxs lack any EGF-like regions and only harbor the sixth extracellular LNS domain, splicing in N-terminal α-Nrx sequences following this LNS6 domain. The truncated proteins encoded by γ-Nrx transcripts lack extracellular LNS or EGF-like domains but retain the transmembrane and intracellular tail domains [8] (Figure 1).

Intracellular Nrx domains recruit cytoskeletal proteins and molecular scaffolds such as calcium/calmodulin-dependent serine protein kinase (CASK), Mints, and protein 4.1, thereby tethering them to the presynaptic machinery [9]. It is possible that these interactions mediate intracellular signaling, an exciting possibility that remains to be explored. The extracellular domains of these Nrxs can orchestrate discrete synaptic signaling pathways by interacting with several structurally unrelated extracellular binding partners, such as the secreted protein cerebellins (Cbls), neurexophilins, the transmembrane proteins neuroligins (Nlgs), α-dystroglycan, leucine-rich repeat transmembrane proteins (LRRTMs), calsyntenin 3, and receptor-type molecules such as the gamma-aminobutyric acid receptor (GABAR) and latrophilins, thereby performing distinct roles via diverse signaling pathways [10]. Collectively, Nrxs govern the specificity of trans-synaptic bridges due to them physically connecting the presynaptic and postsynaptic machinery.

The alternative splicing of six canonical sites (SS1–SS6) can also contribute to increasing Nrx functional and structural diversity, with SS1, SS2, SS3, and SS6 being present only within α-Nrxs, while SS4 and SS5 are found in both α- and β-Nrxs [11]. Through these different combinations of alternative promoter and alternative splice site utilization, mice can theoretically express over 12,000 Nrx transcript isoforms, potentially allowing for the dynamic fine-tuning of the binding affinities of these Nrxs for a range of targets while also contributing to high levels of overall synaptic structural diversity [9]. For example, activity-dependent changes in Nrx SS4 alternative splicing can markedly alter the binding of these proteins with Cbls and LRRTMs, by which SS4–Nrxs bind to LRRTMs, whereas SS4+ Nrxs bind to Cbls [12].

### 2.2. The Structural Diversity of Neuroligins

Nlgs are postsynaptic CAMs that, in humans, are encoded by five genes (*NLG1*, *NLG2*, *NLG3*, *NLG4*, and *NLG4Y*), while, in mice, they are encoded by four genes (*Nlg1–4*) [13,14]. Full-length Nlgs are composed of an N-terminal domain, an extracellular globular cholinesterase-like domain, a highly O-glycosylated stalk domain, a single-pass transmembrane helical domain, and a short C-terminal cytoplasmic tail terminating in sites for PDZ domain-binding [15] (Figure 1). The extracellular acetylcholinesterase-like domain of Nlgs enables them to interact with presynaptic Nrxs in an activity-dependent fashion, while the C-terminal domain of Nlgs can bind to the PDZ domains of postsynaptic scaffold proteins, including PSD-95 and gephyrin, which are involved in the anchoring of functional surface receptors and signaling proteins [16]. The differences in the postsynaptic scaffold proteins with which Nlgs interact suggest that different members of this family may play distinct roles in the context of synaptic function.

Two alternative splice sites (splice site A and splice site B) also contribute to the generation of different Nlg transcript isoforms [15]. These splice sites are located within the cholinesterase-homologous region responsible for interaction with Nrxs, suggesting that the alternative splicing of Nlgs may dynamically impact their interaction with Nrxs. For example, Nlg-1 splice site B insertion can interfere with the ability of the resultant protein to interact with α-Nrxs and reduce β-Nrxs binding [17]. In contrast, a version of Nlg-1 that lacks splice site B induces the removal of the N-linked glycosylation site within splice site B and subsequently recruits more α-Nrxs [18]. Moreover, Nlg-1 splice site A insertion can enhance heparan sulfate (HS)-binding affinity, which is important for Nrx interaction, thus bolstering the recruitment of Nrxs [19].

## 3. The Neurexin–Neuroligin Mediated Trans-Synaptic Modulation

### 3.1. The Dynamic Synaptic Regulation by Neurexins

Although it has been a focus of extensive research interest for over two decades, the precise functional roles played by Nrxs remain poorly understood. Early evidence suggested that these Nrxs and their binding partners can modulate synapse numbers and distributions, thereby contributing to the recruitment of pre-and postsynaptic machinery [8,20]. These findings, together with data from recent studies, strongly suggest that Nrxs are key regulators of the overall functionality of synapses, shaping synaptic processes such as transmission and plasticity [8,21].

Several mechanisms govern the roles of Nrxs in organizing diverse synaptic properties. Primarily, Nrxs influence components of the presynaptic machinery and synaptic functional efficiency. For example, Nrx deletion results in the loss of presynaptic active-zone GABARs, decreasing the sensitivity of neurotransmitter release to GABAR activation at both excitatory and inhibitory synapses [22]. Given the importance of presynaptic GABARs in the nucleation of signaling complexes that control the release of neurotransmitters, the ability of Nrxs to regulate these receptors enables them to further govern synaptic transmission and plasticity from the presynaptic perspective [23,24,25]. Additionally, conditional knockout mice deficient for all three β-Nrxs exhibit impaired presynaptic release probability. However, this phenotype is not due to a direct presynaptic effect but is attributed to a trans-synaptic regulatory loop in which presynaptic β-Nrxs regulate postsynaptic tonic endocannabinoid signaling [26].

The critical roles played by Nrxs in the synapse are not limited to presynaptic modulation. Nrxs can also orchestrate postsynaptic properties, thereby shaping the input/output relations of their resident trans-synaptic circuits. For instance, the expression levels of the N-methyl-D-aspartate receptor (NMDAR) and the α-amino-3-hydroxy-5-methyl-4-isoxazolepropionic acid receptor (AMPAR) are affected by the constitutive inclusion of SS4 in hippocampal presynaptic Nrx-1 and Nrx-3, respectively [27]. Accordingly, conditional control of insertions at Nrx-3 SS4 suppresses the responsivity of glutamate receptor responses mediated by AMPARs, whereas the same manipulation in Nrx-1 enhances NMDAR-mediated glutamate receptor postsynaptic strength. Collectively, these findings suggest an economical molecular mechanism whereby Nrxs and their alternative splicing can contribute to the regulation of AMPAR- and NMDAR-mediated excitatory postsynaptic strength and plasticity [28].

Likewise, GABARs require inhibitory presynaptic terminals for their postsynaptic localization. Specifically, ectopic expression of Nrx-3 in presynaptic neurons could recruit GABARs to postsynaptic sites, thus establishing a trans-synaptic interaction [29]. Consistent with this finding, the Nrx-3 knockout shows altered inhibitory postsynaptic strength, with a pronounced impact on inhibitory postsynaptic current (IPSC) amplitudes in males. In contrast, the same change results in enhanced IPSC amplitudes in females [30]. Moreover, circumstantial evidence indicates that GABAergic synapse specification is influenced by the expression of the highly selective Nrxs SS4 splicing factor sam68-like molecule 2 [31]. These data link the Nrx expression levels and their alternative splicing modulations with the functional regulation of excitatory and inhibitory postsynaptic transmission and plasticity.

Another important mechanism that deserves special attention in relation to Nrxs in synaptic efficiency is the process of proteolytic cleavage. Physiologically, Nrx-1 is cleaved by a disintegrin and metalloproteinase-10 (ADAM-10), resulting in 4–6% of Nrx-1 in the adult brain existing as a soluble ectodomain protein. Blocking ADAM10-mediated Nrx-1 cleavage dramatically increases the synaptic Nrx-1 content, thereby elevating the percentage of excitatory synapses containing Nrx-1 nanoclusters [32]. In fact, the ectodomain cleavage might be critical for the synaptic activity mediated by Nrxs since blocking ectodomain cleavage by metalloproteases reduces β-Nrxs mobility and enhances glutamate release [33]. Moreover, a loss of presenilins that mediate Nrx cleavage induces a drastic accumulation of Nrx C-terminal fragments in cultured rat hippocampal neurons and mouse brains, which coincides with synaptic and memory impairments. These findings suggest that impaired Nrx proteolytic processing may be an early event in the development of dysfunctional synaptic plasticity [34].

Glycosylation of Nrxs may also contribute to their functions in synaptic transmission and plasticity. Nrxs are HS proteoglycans, and the HS component plays a critical role in high-affinity Nrx interaction with Nlgs, LRRTMs, and novel ligands. In line with this notion, reductions in the frequency and amplitude of miniature excitatory postsynaptic currents (EPSCs) recorded from hippocampal neurons have been reported in mice harboring a mutation that interferes with Nrx-1 HS modification, indicating that the HS modification is required for the regulation of synaptic transmission [35]. Given that HS modification of Nrxs is tightly regulated by an activity-dependent mechanism, further research aimed at clarifying how Nrx glycosylation participates in synaptic functional modulation is necessary to shine additional light on this dynamic regulatory process. Overall, the available evidence is consistent with a model in which Nrxs exhibit a high degree of plasticity in the context of synaptic activation, allowing them to shape synaptic transmission and plasticity.

### 3.2. The Dynamic Synaptic Regulation by Neuroligins

Nlgs aid in organizing and orchestrating several aspects of synaptic function [36]. While early experimental evidence suggests that Nlgs can induce the recruitment of presynaptic specializations in an Nrx-dependent manner, further analyses of mice constitutively lacking Nlg1/2/3 expression suggest that these Nlgs are dispensable in the context of initial synaptic formation [37]. Instead, Nlgs appear to serve as mediators of synaptic transmission and plasticity that are modulated by neural activity, leading to activity-induced synaptic circuit functional reshaping, as emphasized in several recent studies [38].

Firstly, the expression profiles of Nlgs are highly specific, with Nlg-1 localizing mainly to excitatory synapses, Nlg-2 localizing primarily to inhibitory synapses, Nlg-3 localizing to both of these synaptic types, and Nlg-4 localizing primarily to the glycinergic synapses of the retinal system [39]. In agreement with their subcellular distributions, Nlg isoforms contribute differently to the function of glutamatergic vs. GABAergic synapses through their capacity to assemble appropriate scaffolds and functional receptors in the postsynaptic membrane opposing the presynaptic terminals. Specifically, Nlg-1 favors the functional modulation of glutamatergic synapses by recruiting NMDARs via the PSD-95 scaffold proteins and trapping surface-diffusing AMPARs by binding with PSD-95 and stargazin [40]. In contrast, Nlg-2 recruits GABARs or glycine receptors through a specific interaction with gephyrin, driving the functional properties of inhibitory synapses [41]. Thus, Nlgs are confined to excitatory synapses or inhibitory synapses, positioning them to influence the excitation/inhibition ratio, the imbalance of which leads to synaptic dysfunction-associated pathologies (Figure 2A). Indeed, manipulating Nlg expression levels in vitro and in vivo has demonstrated their isoform-specific modulation of synaptic functions. The overexpression of Nlg-1 can specifically enhance AMPAR- and NMDAR-mediated EPSCs in an NMDAR-dependent fashion. In contrast, Nlg-1-knockout mice exhibit reduced NMDAR-mediated EPSC amplitudes and a loss of NMDAR-dependent long-term potential (LTP) [42,43,44]. Of note, NMDAR-dependent LTP is not dependent on the binding of Nlg-1 to PDZ-domain-containing proteins, such as PSD-95. Rather, it requires Nlg-1 binding to Nrxs, as the rescue of LTP by Nlg-1 can be prevented by the mutation of residues critical for Nrx binding. This finding demonstrates the specific and pronounced regulatory role played by Nlg-1 in the context of excitatory synaptic transmission in an Nrx-dependent manner. However, the rescue of basal NMDAR-mediated synaptic transmission after Nlg deletion requires the Nlg-1 intracellular domain but not Nrx binding [45]. This molecular dissociation of Nlg-1 domains required for LTP versus those required for the maintenance of basal NMDAR-mediated synaptic transmission indicates the complexity of the molecular architecture responsible for regulating synaptic strength at excitatory synapses.

Research on Nlg-2 has fundamentally shaped our understanding of the molecular architecture of Nlg-2 as a central organizer of inhibitory synapses. The protein components of the GABAergic postsynaptic complex, GABARs and gephyrin, are reduced in Nlg-2-knockout mice [41], accompanied by a general reduction in inhibitory synaptic transmission. In contrast, overexpression of Nlg-2 reportedly results in a specific increase in IPSC amplitude, underscoring the role of Nlg-2 as a mediator and regulator of inhibitory synaptic transmission [46,47].

Nlg-3 can either upregulate or downregulate inhibitory synaptic transmission in a splice-variant-dependent manner, suggesting that the specific subcellular localization of the Nlg-3 isoforms may contribute to the functional differences observed between them [48,49]. However, when overexpressed, Nlg-3 enhances excitatory transmission and presynaptic vesicular glutamate transporter 1 expression, irrespective of the Nlg-3 splice variant. Together, these data indicate that alterations in Nlg expression can modulate both excitatory and inhibitory synaptic transmission, thus contributing to synaptic plasticity.

It is also noteworthy that Nlgs function as mediators of synaptic transmission and plasticity in a phosphorylation-dependent manner. For example, calcium/calmodulin-dependent protein kinase II (CaMKII)-mediated phosphorylation of T739 in the Nlg-1 C-terminal domain results in increased surface trafficking, while the mutant protein that could not be phosphorylated leads to reduced Nlg-mediated excitatory synaptic potentiation [50]. Nlg phosphorylation also plays a role in governing postsynaptic protein recruitment, including both surface receptors and scaffold proteins. For instance, protein kinase A can phosphorylate Nlg-1 at S839 and dynamically regulate PSD-95 binding since a phosphomimetic mutation of Nlg-1 at S839 significantly reduces PSD-95 binding [51]. Nevertheless, Nlg-1 phosphorylation at Y782 inhibits gephyrin binding rather than promoting PSD-95 recruitment [52]. Proline-directed Nlg-2 phosphorylation at the S714 residue can ablate gephyrin binding, even though the residue does not fall in the gephyrin-binding domain. A previous study shows that S714 phosphorylation promotes the recruitment of the peptidyl-proly cis-trans isomerase (Pin1) enzyme, which mediates Nlg-2 isomerization and disrupts gephyrin interaction [53].

Nlg phosphorylation also influences the recruitment of functional surface receptors, with phosphorylation of Nlg-1 at Y782 recruiting cell-surface AMPARs and thus augmenting AMPAR-mediated EPSCs [54]. On the other hand, endogenous Nlg-2 phosphorylation is associated with reductions in spontaneous GABAR-mediated postsynaptic current amplitudes due to reduced GABAR and gephyrin retention at inhibitory synapses [53]. As Nlg phosphorylation is controlled in an activity-dependent fashion, this may represent another distinct mechanism whereby these proteins can dynamically modulate synaptic transmission and plasticity.

Indeed, Nlgs have been found to undergo activity-dependent proteolytic cleavage, which can alter their activity and/or expression. For example, proteolytic cleavage of Nlg-1 prevents its surface expression, leading to the destabilization of trans-synaptic Nrx–Nlg complexes [55,56]. Furthermore, the extracellular fragments of Nlg-1 generated through its proteolytic cleavage can bind to presynaptic metabotropic glutamate receptors, ultimately leading to their activation and the suppression of glutamate release, thereby reducing synaptic strength [57]. Likewise, the induction of Nlg-3 proteolytic cleavage by protein kinase C results in reduced synaptic strength that can be counteracted by the prevention of cleavage. It should be noted that the proteolytic cleavage of Nlg-1 and Nlg-2, indeed, depends on the presence of Nlg-3 through Nlg-1/Nlg-3 or Nlg-2/Nlg-3 heterodimers, indicating the potential role of Nlg-3 as a key regulator of other Nlg cleavage events [58].

## 4. The Regulators of Neurexin–Neuroligin Interaction

The mechanistic importance of the Nrx–Nlg trans-synaptic interaction in the regulation of synaptic functionality is also supported by the ability of the two proteins to bind a variety of other proteins that modulate their interaction in the synaptic cleft. These proteins include hevin (high endothelial venule protein), SPARC (secreted protein acidic and rich in cysteine), and MAM domain-containing glycosylphosphatidylinositol (GPI) anchor proteins (MDGAs).

### 4.1. Hevin

Hevin, also known as SPARC-like protein 1, is a member of the SPARC family that is highly expressed in developing astrocytes, even after reaching adulthood [59]. Structurally, hevin is a cysteine-rich glycoprotein containing a flexible acidic N-terminal region and a globular C-terminal region, which contains a follistatin-like (FS) domain and an extracellular calcium-binding (EC) domain. The FS–EC tandem domains exist as a monomer in solution, maintaining an elongated structure where the FS and EC domains do not interact, suggesting their independent functions. Further studies using surface plasmon resonance have shown that the FS domain contains residues responsible for the hevin–Nlg interaction. It has also been found that the FS domain is sufficient for Nrx binding and that the interaction is calcium-dependent [60].

Interestingly, these structural findings correspond with previous results showing that hevin forms a trans-synaptic bridge between Nrx-1α and Nlg-1, which otherwise do not directly interact with one another under physiological conditions, thus regulating the formation and refinement of thalamocortical glutamatergic synapses [61]. It is found that expression of Nlg-1, PSD-95, homer-1, and the NMDAR subunits are significantly reduced in the cortical postsynaptic membranes of hevin-knockout mice compared with wild-type mice [61]. Of note, in addition to the recruitment of NMDARs, the treatment of cortical neurons with hevin can also raise the number of AMPARs on the cell surface, increasing both the amplitude and frequency of AMPAR-mediated EPSCs [62]. Furthermore, increased co-localization of hevin with the excitatory synaptic markers vesicular glutamate transporter 1, AMPAR subunit, and NMDAR subunit is observed in epileptogenesis, while no co-localization is seen with the inhibitory synaptic markers vesicular GABA transporter and GABAR subunit, suggesting an association of hevin with the modulation of excitatory synapses [63]. Therefore, the trans-synaptic interaction mediated by hevin might be exploited to modulate synaptic reorganization under various neurological conditions via the stabilization of the Nrx–Nlg trans-synaptic bridges.

### 4.2. SPARC

SPARC, also called osteonectin, is a prototypical member of the SPARC family of biologically active glycoproteins expressed largely in microglia and some subcortical astrocytes in the central nervous system [64]. Examination of the protein structure shows that SPARC is highly homologous to hevin, with three identical structural domains, including the N-terminal, FS, and EC domains [65]. Notably, the SPARC FS domain shares 56% sequence identity with the FS domain of hevin, and they are structurally similar. This intriguing feature suggests that SPARC may interact directly with Nrxs and Nlgs, and, indeed, it has been recently demonstrated that SPARC can bind both Nrxs and Nlgs with a similar affinity as hevin. In particular, SPARC resembles hevin in binding Nlg-2 with comparable affinity and in binding Nrx-1α in a similar calcium-dependent manner, suggesting that it may compete with hevin for binding to both Nrxs and Nlgs [60]. In short, both SPARC and hevin can stabilize the Nrx–Nlg trans-synaptic bridge, and the ratio of SPARC versus hevin regulates Nrx–Nlg interaction and determines the net effect on synaptic function.

It is possible that the actions of one regulator might oppose those of another. Characteristically, SPARC antagonizes the action of hevin in synaptogenesis, with SPARC specifically inhibiting hevin-induced excitatory synaptogenesis; SPARC-null mice show increased synapse formation [65]. Strikingly, SPARC may play multiple roles in synaptic function by regulating the postsynaptic glutamate receptors. During development, SPARC acts more like a “molecular brake” to prevent the over-accumulation of AMPARs, thereby altering both the amplitude and frequency of AMPAR-mediated mEPSCs and decreasing synaptic strength [66]. However, under injury or disease in the mature nervous system, SPARC expression is increased, accompanied by the upregulation of AMPAR substrates at synapses and enhanced synaptic function [67]. Specifically, SPARC treatment reduces the loss of AMPAR subunits in the synapses of hippocampal neurons following oxygen-glucose deprivation, and SPARC may play a novel role in regulating neuronal health and recovery following central nervous system injury. In addition, pretreatment with SPARC is accompanied by a significant increase in synaptic NDMAR subunit levels under the same condition, which could, in turn, enhance synaptic strength and plasticity [67]. Taken together, it is conceivable that SPARC has differential effects on synaptic signaling pathways during development and following injury/disease. It would be important to test the effect of SPARC on the trans-synaptic Nrx–Nlg connection and determine whether this trans-synaptic bridge influences the activities of functional receptors.

### 4.3. MDGA

MDGAs are vertebrate-specific proteins belonging to the immunoglobulin superfamily and are specifically expressed in the nervous system. The two homologous MDGA proteins, MDGA1 and MDGA2, have a characteristic domain organization composed of an N-terminal signal peptide followed by six immunoglobulin domains, a MAM domain, and a C-terminal domain containing a cleavage site for GPI for anchorage in the cell membrane [68]. Structural determination of the MDGA extracellular region shows a compact, approximately triangular structure that is stabilized by extensive interdomain contacts [69].

Unlike hevin and SPARC, MDGAs interact with Nlgs and, thereby, block sites for Nrx binding, thus suppressing trans-synaptic bridge formation and disrupting synaptic activity [69,70]. The affinity profiles of different MDGAs are Nlg isoform-specific, providing unique modes for regulating different neuronal populations during synaptic development and transmission. Specifically, MDGAs bind in a cis configuration with a lower affinity to Nlg-3 and Nlg-4 than to Nlg-1 and Nlg-2. In addition, MDGA1 selectively forms complexes with Nlg-2, and MDGA2 selectively forms complexes with Nlg-1 [71]. Analysis of mouse mutants shows that targeted mutations of MDGA1 and MDGA2 promote the activities of inhibitory and excitatory synapses, respectively, suggesting functional divergence between the proteins [69]. In line with this notion, RNAi-mediated knockdown of MDGA1 increases the number of inhibitory synapses in cultured rat hippocampal neurons without affecting the number of excitatory synapses [72]. Mutation of MDGA2 increases excitatory synapse numbers and elevates excitatory transmission by blocking Nlg-1 interaction with Nrxs, with no influence on inhibitory synapses [73]. Intriguingly, MDGAs and hevin compete for the same Nlg binding sites and such competition may shape the excitatory and inhibitory balance mediated by molecular crosstalk between different modifiers of the Nrx–Nlg interaction [60]. Thus, examining how different MDGAs modulate Nrx–Nlg interaction will enhance our understanding of the mechanisms underlying synapse plasticity and provide new insights into the etiology of neurological disorders.

## 5. The Role of Trans-Synaptic Neurexin–Neuroligin Interaction in Pathological Pain

Given that trans-synaptic transmission and plasticity are closely associated with the pathophysiology of many neurological disorders, it is likely that Nrx–Nlg-mediated trans-synaptic interaction may also play a role in pathological pain. Indeed, several studies have suggested that Nrxs and Nlgs play vital roles in the context of pathological pain [8,9]. Thus, determining the underlying mechanisms may assist the development of new tools to manipulate trans-synaptic connections and the identification of new analgesic targets.

### 5.1. Disrupting Trans-Synaptic Transmission of Nociceptive Signals by Targeting Neurexins

Nrx-2 upregulation is evident in the spinal cord dorsal horn in rats after injection of complete Freund’s adjuvant (CFA) into the hind paw (Table 1). Importantly, the silencing of Nrx-2 attenuates CFA-induced inflammatory mechanical and heat hyperalgesia, which is associated with a decreased AMPAR expression in the spinal dorsal horn [74]. In addition, the disruption of trans-synaptic interaction between Nrx-1 and Nlg-1 using an Nrx-1 Fc chimera reduces spinal nerve ligation (SNL)-induced mechanical hypersensitivity, in part by disrupting spinal Nlg-1/PSD-95/NMDAR signaling. These findings suggest that Nrxs may contribute to pain hypersensitivity by interacting with Nlgs, leading to the subsequent activation of downstream signaling cascades in the spinal dorsal cord [8]. Gabapentinoid drugs, which are a cornerstone in the treatment of neuropathic pain, may function in large part through the modulation of Nrx-1α [75]. Genomic analyses have revealed that migraine susceptibility is tied to bidirectional *NRXN2–CASK* gene interactions [76]. While many factors may contribute to genetic interactions, these findings provide new insights regarding the molecular basis of pathological pain and underscore the relevance of Nrxs as potential new targets for pharmacological pain intervention.

### 5.2. Disrupting Trans-Synaptic Transmission of Nociceptive Signals by Targeting Neuroligins

Many studies have provided strong evidence for the role of Nlgs as mediators of pathological pain. Following intraplantar CFA injection, synaptic Nlg-1 concentrations are observed to rise significantly in the spinal cord dorsal horn. Silencing of Nlg-1 expression with a siRNA construct is sufficient to suppress synaptic NMDAR subunit expression and attenuate inflammatory mechanical and thermal hypersensitivity [77]. In a postoperative pain model, Nlg-1 expression is significantly increased in the ipsilateral dorsal horn. In contrast, spinal knockdown of Nlg-1 expression is sufficient to alleviate incision-induced pain through interference with the Nlg-1/PSD-95 interaction and synaptic AMPAR subunit targeting [78].

Intriguingly, the expression of Nlg-2, which is normally localized primarily to inhibitory synapses, is upregulated in the spinal dorsal horn after SNL in rats, with corresponding increases in the co-localization and interaction of Nlg-2 and PSD-95. Meanwhile, siRNA-mediated Nlg-2 knockdown in the spinal cord disrupts these interactions and reduces pain [79]. In addition to neuropathic pain, the role of Nlg-2 as a regulator of postoperative pain is also reported. There is an increase in postsynaptic Nlg-2 membrane expression in the ipsilateral spinal dorsal horn after plantar incision, and intrathecal siRNA-mediated Nlg-2 knockdown prior to injury is sufficient to interfere with the AMPAR subunit targeting the postsynaptic membrane and attenuates postoperative pain [80]. Thus, it is possible that the the Nlg-2 protein may undergo a functional shift from inhibition toward excitation in the context of pathological pain, favoring more excitatory signaling (Figure 2B). Additional studies will be needed to determine whether the phosphorylation and proteolytic cleavage of Nlgs can also influence pain modulation since these regulatory activities are closely associated with synaptic functionality. Moreover, their mediators, such as CaMKII and metalloproteinases, are also closely related to pain regulation [85,86,87].

### 5.3. Disrupting Trans-Synaptic Transmission of Nociceptive Signals by Targeting Regulators of the Neurexin–Neuroligin Interaction

Several secreted proteins, especially those regulating the interactions between Nrxs and postsynaptic Nlgs, may be regulators of pathological pain. For example, hevin-containing sensory neurons are found in the human sensory ganglion, indicating that hevin may participate in nociceptive transmission [88]. In addition, hevin expression and AMPAR membrane trafficking are increased in a murine model of remifentanil-induced postoperative pain. Knocking down spinal hevin expression is sufficient to attenuate such hyperalgesia and to reduce AMPAR substrate membrane localization following remifentanil treatment, emphasizing the direct regulatory roles played by hevin in pain [81]. Hevin is also required for the maintenance of neuropathic pain since the neutralization of secreted hevin with a monoclonal antibody effectively alleviates neuropathic pain. Mechanistically, hevin acts as a pro-nociceptive mediator, in part through NMDAR-mediated signaling and NMDA-induced mechanical allodynia, and inward currents in spinal cord lamina II neurons are reduced in hevin-null mice [82].

Modulating SPARC protein expression may represent a novel therapeutic opportunity for the treatment of pathological pain. SPARC-null mice exhibit chronic lower back and radicular pain-like behavior and have traditionally been used as a model of chronic back pain [83]. Consistently, low back pain patients commonly show increased methylation of the SPARC promoter, which silences SPARC activity. These findings suggest that negative epigenetic and post-translational regulation of SPARC may underlie the pathogenesis of pain [84]. In light of the ever-increasing identification of regulators associated with the Nrx–Nlg interaction in pathological pain, in-depth research is warranted to delineate the specific roles of these regulators and their associated trans-synaptic signaling pathways in pathological pain.

A recent study shows that MDGA1 mutations may contribute to cognitive deficits due to altered synaptic transmission and plasticity, including reduced suppression of inhibitory synapse development and compromised hippocampal LTP [89]. The mutation of MDGA2 contributes to the altered hippocampal LTP, with elevated excitatory transmission and suppressed excitatory synapse development [73]. Given that pain and cognition are known to interact reciprocally and share common neural substrates, especially those that contribute to the synaptic transmission and plasticity, the synthesis of current research findings regarding MDGAs in cognitive function has implications for the treatment and management of pain conditions [90,91]. Furthermore, MDGAs are differentially expressed by subpopulations of neurons in both the central and peripheral nervous systems, including neurons of the cerebral cortex, spinal cord, dorsal root ganglion, and trigeminal ganglia [68]. Collectively, upon the Nrx–Nlg interaction exerting critical action in nociceptive transmission, further investigation is required to fully elucidate the modulated mechanisms involved and enable the development of improved treatment strategies for pain by targeting MDGAs.

Together, previous studies have clearly demonstrated that Nrxs and Nlgs are critical modulators of pathological pain development. Future research is warranted to fully elucidate the underlying mechanisms by which the regulators influence Nrx–Nlg mediated trans-synaptic signaling pathways to gain insights and discover new therapeutic targets for pathological pain.

## 6. Concluding Remarks and Perspectives

The Nrx–Nlg interaction serves as a central player in trans-synaptic transmission and synaptic plasticity. The interactions between these proteins are not stationary but are continuously remodeled through activity-dependent changes, suggesting the presence of a cell-specific molecular code for synaptic efficiency. Trans-synaptic adhesion mechanisms thus enable the Nrx–Nlg interaction to play an integral role as a mediator of pathological pain. The current evidence indicates the mechanisms by which regulatory proteins interacting with Nrx–Nlg cooperate to shape trans-synaptic bridges and their roles in the development and resolution of pathological pain. Nevertheless, deciphering the precise molecular logic guiding therapeutic targets remains a formidable but rewarding challenge that will consume the research in the field for years to come.

## Figures and Tables

**Figure 1 cells-11-01940-f001:**
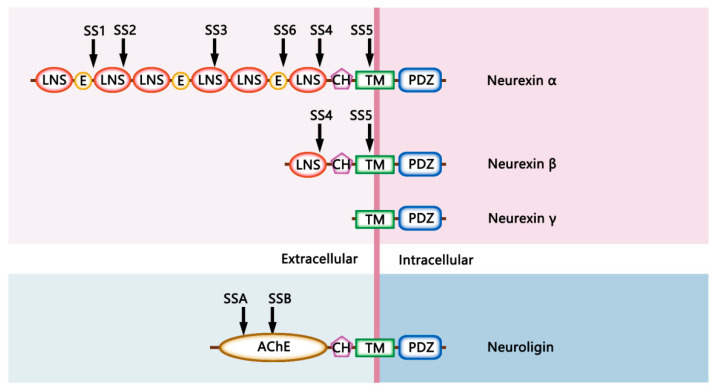
Putative structure of neurexins and neuroligins. Neurexin α consists of six laminin/neurexin/sex-hormone (LNS) globulin domains, three interspersed extracellular epidermal growth factor (EGF)-like repeats, a rigid and highly O-linked glycosylated (CH) stalk, a transmembrane (TM) domain, and a C-terminal PDZ binding motif. Neurexin β lacks any EGF-like regions and only harbors the sixth extracellular LNS domain. Neurexin γ lacks extracellular LNS or EGF-like domains but retains the transmembrane and intracellular tail domains. Full-length Neuroligins are composed of an N-terminal domain, an extracellular globular acetylcholinesterase-like domain (AChE), a CH stalk, a TM domain, and a C-terminal PDZ binding motif. Alternative splice insert sites are indicated as they are referred to in the text.

**Figure 2 cells-11-01940-f002:**
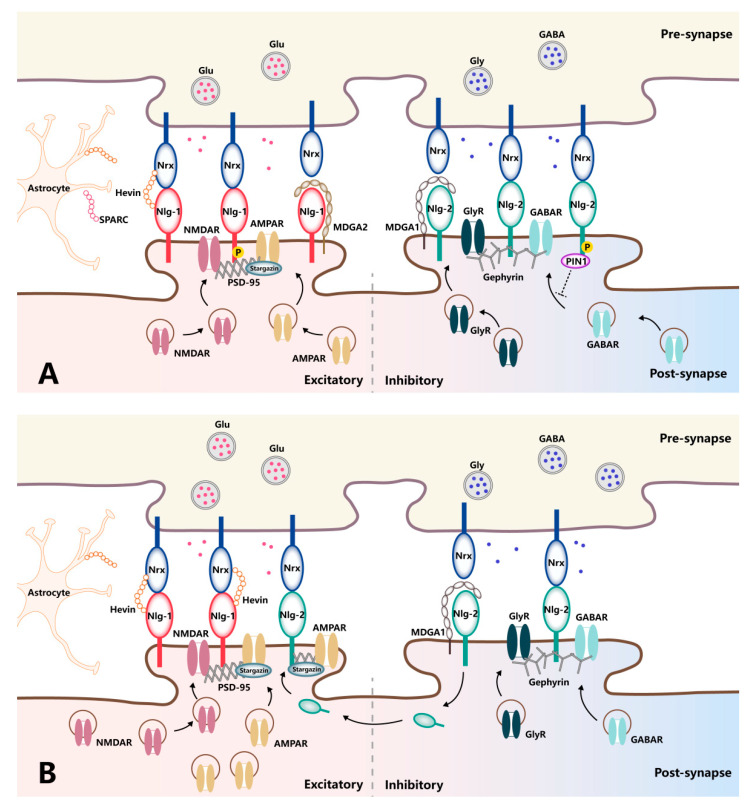
Schematic diagram of trans-synaptic interaction between neurexins and neuroligins in normal and pathologic pain. (**A**) In normal conditions, Nlg-1 and Nlg-2 contribute differently to the function of excitatory and inhibitory transmissions, respectively, in front of corresponding presynaptic Nrxs. Nlg-1 favors the functional modulation of glutamatergic synapses by recruiting NMDARs via the PSD-95 scaffold and trapping surface-diffusing AMPARs by binding with PSD-95 and stargazing. Nlg-2 recruits GABARs or glycine receptors through a specific interaction with gephyrin. The mechanistic importance of Nrx–Nlg is associated with a range of proteins capable of modulating their interaction at the synaptic cleft, in which hevin and SPARC directly interact with Nrx and Nlg, while MDGAs occupy the interaction site between Nrx and Nlg and, thereby, block Nrx–Nlg interaction. (**B**) In pathologic pain, Nlg-1, as well as the recruitment of corresponding excitatory glutamatergic receptors, is upregulated in the context of pathological pain. Meanwhile, traditionally inhibitory Nlg-2 undergoes a fundamental shift in functionality from inhibition towards excitation, with an increase in co-localization with PSD-95 and subsequent AMPAR subunit targeting under the same condition.

**Table 1 cells-11-01940-t001:** Preclinical evidence regarding the role of trans-synaptic cell-adhesion molecules in pathological pain. CFA: complete Freund’s adjuvant, SNL: spinal nerve ligation, CCI: chronic constriction injury, AMPAR: α-amino-3-hydroxy-5-methyl-4-isoxazolepropionic acid receptor, NMDAR: *N*-methyl-d-aspartate receptor.

Molecular Names	Type of Pain	Animals	Functional Receptors	Key Reference
Nrx-1	SNL	Rats	NMDAR	Lin et al., 2015 [8]
Nrx-2	CFA	Rats	AMPAR	Xu L et al., 2020 [74]
Nlg-1	CFA	Mice/Rats	NMDAR	Zhao et al., 2018 [77]
Nlg-1	Postoperative pain	Rats	AMPAR	Guo et al., 2018 [78]
Nlg-2	SNL	Rats	NA	Dolique et al., 2013 [79]
Nlg-2	Postoperative pain	Rats	AMPAR	Guo et al., 2018 [80]
Hevin	Postoperative pain	Murine	AMPAR	Wang et al., 2020 [81]
Hevin	Inflammation pain/CCI	Mice	NMDAR	Chen et al., 2022 [82]
SPARC	Chronic back pain	Mice	NA	Lee et al., 2022 [83]
SPARC	Chronic back pain	Human	NA	Tajerian et al., 2011 [84]

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
