# Peer review of "Modulation of Trans-Synaptic Neurexin–Neuroligin Interaction in Pathological Pain"

_cells, 2022, doi:10.3390/cells11121940_

Round 1

Reviewer 1 Report

This is a well written and organized review of neurexins, neuroligins, their molecular partners, and their potential role in pathological pain treatment.  The authors review the structure and function of neurexins and neuroligins, their molecular interactions, and the literature regarding their role in pain modulation.  Although some of this material has been reviewed extensively in other contexts, the inclusion of the role of these factors in pain modulation has not been as extensively covered.  As such, this is a useful addition to the literature that summarizes the state of the field.  I have a few minor suggestions for the manuscript.

First, the title promises a bit more than the article actually delivers.  I would suggest softening slightly.

For Figure 1, it would help to label the 'extracellular' and 'intracellular' compartments.

The writing is very good overall.  However, there are some alterations in tense that affect the readability, i.e. switching from past to present tense, especially starting at section 3.  I would suggest writing the entire review in the present tense.

Also, this is a bit of a stylistic comment and pet peeve, but there are repeated uses of phrases like "Previous work has shown that...", which are superfluous and overused in scientific writing.  It is implied that the findings discussed in the review are from previous work, so it is sufficient to just start a sentence stating these findings. 

Section 5 is the heart of the review and it does a nice job of reviewing the past literature on the role of these molecules in pain modulation.  However, it doesn't rise much beyond rehashing the results of past studies and could benefit from an attempt to synthesize these results more succinctly into a framework for future studies.  Perhaps, adding an additional figure here would help unify this section?

Reviewer 2 Report

Re: Modulation of Trans-Synaptic Neurexin-Neuroligin Interaction for Pathological Pain Treatment

This is an interesting review. I enjoyed reading it and I now feel as I am up to date on the topic. The relationship in modulating pain can in time be a focus for therapeutic targets.

The review is very extensive and appears to cover the filed very well. I think it will be a good review for anyone studying the topic or who would like to learn what has been covered to date to then design potentially the next steps in further research.

It also appears that the citations are well represented throughout, and the figure helped as a visual aid.

I did not see the need for any changes or additions.

Author Response

Thanks very much for your affirmation to our manuscript. We would like to express our sincere appreciation to Reviewer #2 for the recognition and encouragement to us.

Reviewer 3 Report

The manuscript “Modulation of Trans-Synaptic Neurexin-Neuroligin Interaction for Pathological Pain Treatment” by Li et al is a review article which describes the trans-synaptic cell-adhesion molecules and its role synaptopathies in the context of pathological pain and other neurological disorders. Generally, the subject is of interest and scientifically sound and contains essential contents. This topic is also of importance for treatment of pathological pain. The manuscript has been well organized and written. However, I have some concerns on the paper.

Although the title of this review includes “Pathological Pain Treatment”, this section is minimally described. It may be necessary for the authors to change the title.

The authors concisely discussed the role of trans-synaptic cell-adhesion molecules in pathological pain. However, a figure or table summarizing these findings may be helpful.

In addition to pathological pain, relationships between synaptogenic proteins including Hevin and SPARC and neurological disorders such as epilepsy, ischemia and Alzheimer’s disease are described in section 5. However, this topic should be described as a new section because this topic is nothing to do with pathological pain.
